# Housing Systems Affect Eggshell Lightness and Free Amino Acid Contents of Egg Albumen in Tosa-Jidori Chickens: A Preliminary Research

**DOI:** 10.3390/ani13111837

**Published:** 2023-06-01

**Authors:** Nonoka Kawamura, Masahiro Takaya, Hideaki Hayashi, Tatsuhiko Goto

**Affiliations:** 1Department of Life and Food Sciences, Obihiro University of Agriculture and Veterinary Medicine, Obihiro 080-8555, Hokkaido, Japan; 2Hokkaido Tokachi Area Regional Food Processing Technology Center, Tokachi Foundation, Obihiro 080-2462, Hokkaido, Japan; 3Department of Veterinary Science, School of Veterinary Medicine, Rakuno Gakuen University, Ebetsu 069-8501, Hokkaido, Japan; 4Research Center for Global Agromedicine, Obihiro University of Agriculture and Veterinary Medicine, Obihiro 080-8555, Hokkaido, Japan

**Keywords:** chicken, egg, free amino acid, housing system, stress

## Abstract

**Simple Summary:**

Although there are several housing systems (e.g., cage and litter) for egg layers, there is still no consensus about housing system effects on egg quality traits. Therefore, the purpose of this study is to determine housing effects on egg quality traits, including free amino acids of albumen and yolk. We observed significant housing effects in body weight gain, eggshell weight, yolk weight, eggshell thickness, eggshell lightness, and several albumen amino acids. Staying 7 weeks in a litter condition seemed to be enough to cause a significantly lighter eggshell color regardless of the egg production stages. These results will be important knowledge in the future layer industry.

**Abstract:**

Many countries have gradually shifted to animal welfare-friendly housing systems for egg layers. However, there is still no consensus among researchers on whether the housing system affects egg quality traits. Therefore, this study aimed to determine the effects of housing systems on egg traits and free amino acid contents of albumen and yolk using two types of housing systems, the conventional cage (cage) system and a floor rearing (litter) system. Tosa-jidori (n = 20) hens were divided into two groups. Experiments during the 7 weeks were performed twice by switching the housing systems (first and second stages). One-way analysis of variance was used to evaluate the effects of housing systems on body weight gain, egg traits, albumen and yolk amino acid contents, and fecal corticosterone. We observed significant housing effects in body weight gain, eggshell weight, yolk weight, eggshell thickness, eggshell lightness, and several albumen amino acids (A_Gln, A_His, A_Met, A_Cys, A_Lys, A_Asp, A_Glu, A_Ser, A_Thr, A_Ala, A_Pro, and A_Phe). Notably, a robust effect was seen in eggshell lightness, even after switching housing systems. These results suggest that eggshell lightness and several egg traits, including albumen amino acid contents, can be changed by using the different housing systems.

## 1. Introduction

Eggs can be purchased at low prices, although they are called “complete foods” [1]. Eggs are rich in protein, essential amino acids, and vitamins, except for vitamin C [2]. Free amino acid, one of the ingredients of hens’ eggs, is known to be related to the food taste and functionality after intake. For example, some free amino acids have been shown to have high antioxidant potential [3]. The production of designer eggs has been studied since 1934 [4]. Designer eggs are defined as eggs designed in advance to incorporate the nutrients desired by producers. In recent years, various designer eggs, such as DHA-enriched eggs, have been commercialized in the world [5].

The housing system in layer hens has changed over time. A battery cage (cage) was developed in the 1930s, and the system became the worldwide mainstream of poultry farming in the 1950s [6]. The cage system has been widely used for a long time in the poultry industry because it enables high productivity and maximizes profit under the control of several diseases. However, an enriched cage that has nests, sand, and perch has been used because interest in animal welfare increased in Europe. In 2012, the cage system, except for the enriched cage, was banned by the European Union (EU Directive 1999/74/EC). In addition, egg production under the cage system was also prohibited by law in some States of the United States [7]. In response to these changes in housing systems, many countries in the world have been gradually changing from battery cages to animal welfare-friendly housing systems [8,9]. This change from cage to non-cage housing systems had both favorable and unfavorable consequences in considering welfare and sustainability [10].

The Japanese layer industry produces a large number of eggs, and the production quantity (approximately 2.57 million tonnes based on “hen eggs in shell, fresh” in 2021) is the seventh largest in the world after China, India, the United States, Indonesia, Brazil, and Mexico [11]. Japanese annual per capita consumption of hen’s eggs is the second largest after Mexico [12]. Although there are guidelines for the animal welfare of layers in Japan [13], there is currently no legal restriction. Therefore, it is known that the usage rate of the conventional cage system is still extremely high at over 90% in Japan [13]. The current situation in Japan would be shifted to the world standard housing systems that consider animal welfare in the future [14].

There are many reports that several housing systems for egg layers influenced animal welfare, egg production rate, egg weight, and eggshell quality traits [15,16,17,18,19]. However, there is still no consensus among researchers on whether the housing system affects egg quality in chickens [20]. Our research group has reported that the free amino acid contents of eggs can be altered by both genetic and environmental factors [21,22,23,24,25,26]. Most research studies have selected the conventional cage (cage) system, whereas Nishimura et al. [26] used a floor-rearing (litter) system. These studies investigated breed and feed effects as genetic and environmental factors on egg yolk and albumen amino acid traits, respectively. Until now, there has been a lack of evidence that the housing system itself affects free amino acid traits in eggs. Therefore, egg amino acids were selected as target traits in this study.

Measurement of stress levels has often been used to evaluate animal welfare. Plasma corticosterone is one of the most famous indicators to evaluate acute stress levels in animals. Since measuring plasma corticosterone requires an invasive method, blood sampling itself causes acute stress on animals, which will lead to higher corticosterone levels. Therefore, it will be difficult to assess minor alterations of chronic stress levels changed by the housing systems. To overcome these challenges, previous studies have shown that measuring corticosterone in metabolites (noninvasively collected feces, milk, and eggs) is a good substitute for blood corticosterone [27,28,29,30,31]. In chickens, it is possible to assess objective physiological stress from fecal corticosterone [32].

In this study, we focused on the Tosa-jidori (Japanese Old Type-Tosa), which is one of the Japanese indigenous chicken breeds [33]. This breed is known as one the oldest breeds in Japan and is the smallest breed among Japanese indigenous chickens [33]. In general, the relation of avian egg weight to body weight is known in many species, including Galliformes [24,34]. Since the adult body weight of Tosa-jidori females is around 600 g [33,35], the egg weight is known to be small in proportion to body size. Because of the small body size, we think the Tosa-jidori can be an ideal model breed of chickens. To investigate the effect of the housing system, we adopted switching the housing systems (cage and litter) using the Tosa-jidori model. The main reason for the switching experiment is to determine the effect of the housing systems experimentally. If the housing system has some strong effects on egg quality traits, the significant effects on egg traits will be seen in both the first and second stages (2 months each) after switching the housing systems. 

For these reasons, the purpose of this study was to determine the effects of housing systems on egg traits, free amino acid contents in egg albumen and yolk, and fecal corticosterone using two types of housing systems (cage and litter). We will provide a possibility for making designer eggs using the effect of the housing system on chickens.

## 2. Materials and Methods

### 2.1. Animals

Tosa-jidori (n = 20) hens at the experimental farm of the Obihiro University of Agriculture and Veterinary Medicine, Japan, were reared in group-housing conventional cages with free access to diet and water. Mixed feed for layers (Rankeeper; Marubeni Nisshin Feed Co., Ltd., Tokyo, Japan) was provided, and the ingredients in the mixed feeds have been reported previously [25]. At the age of 17 wks (n = 10) and 15 wks (n = 10), all hens were weighed and allocated to two different housing systems: the conventional cage (cage) system and the floor rearing (litter) system (n = 5 at 17 wks and n = 5 at 15 wks in each group). The lighting cycle was set at 16 h light and 8 h dark. There was no significant difference in body weight between the two groups at the starting point. The mean ± SD of the body weights were 594 ± 86.3 g and 546 ± 55.2 g in cage and litter, respectively. 

### 2.2. Housing Condition and Experimental Design

Two housing systems (cage and litter) were placed side by side within the same facility. Therefore, there were no differences in surrounding environments, including temperature, humidity, and lighting. Since 10 hens were reared in the individual cage (630 cm^2^/hen), the hens could access feed and water but not their feces. There were no perch and nests in the cage system. On the other hand, 10 hens were housed (11.25 m^2^/10 hens) to secure the space for exercising in the litter floor (litter) system, which was similar to the floor rearing system of Nishimura et al. [26]. The litter system was constructed with rice husks as bedding materials with a depth of 20 cm. Therefore, the hens could access the feed, water, rice husks, and feces in the litter system. The litter system included nests and perches. Feed and water were ad libitum in each housing system.

The experimental design is shown in Figure 1. The timing of starting the experiment is set at 0 wks, which is the early laying stage (n = 5 at 17 wks and n = 5 at 15 wks in each group). We defined two production stages as follows: the first stage (0 to 7 wks) and the second stage (7 to 14 wks) (Figure 1). Each stage lasted for 7 weeks until the egg sampling. Before the experiment (0 wks), all the hens were weighed and then divided into two groups (cage and litter). At 6 and 7 wks, all the eggs were collected from each system in the first stage. After collecting eggs at 7 wks, the housing systems were switched to start the second stage. At 13 and 14 wks, all the eggs were collected in the second stage. After collecting eggs at 14 wks, cage and litter hens were kept in the second stage condition until 19 wks. At 19 wks, the litter floor rearing hens were moved to cages for 2 h to sample feces from all hens individually. Feces were individually collected for cage hens as well as the litter hens. The timing of ending the experiment is the middle laying stage (n = 5 at 36 wks and n = 5 at 34 wks in each group). 

### 2.3. Body Weight Gain

Body weight (BW) was measured using an electronic balance (WPB3KOI, AS One Corporation, Osaka, Japan) at 0 wks, 7 wks, and 14 wks during the experiment (Figure 1). The body weight gain (BWG) at first and second stages was calculated by post-BW (7 wks and 14 wks) minus pre-BW (0 wks and 7 wks), respectively.

### 2.4. Egg Production Rate

Daily egg number was counted in cage and litter systems. Since each housing system has 10 hens, the maximum daily egg production is 10 eggs. The egg production rate (%) during 1 week was calculated through the experimental period (from 0 wks to 19 wks).

### 2.5. Egg Traits

All eggs were collected daily and stored at room temperature. All the traits from both cage and litter systems were collected under the same schedule. A total of 10 egg traits were measured using 79 eggs (n = 20 at 6 wks, n = 20 at 7 wks, n = 20 at 13 wks, and n = 19 at 14 wks). Egg weight (EW), yolk weight (YW), albumen weight (AW), and eggshell weight (SW) were measured using an electronic balance (EK-6000H; A&D Company, Ltd., Tokyo, Japan). The length of the long axis of the egg (LLE) and the length of the short axis of the egg (LSE) were measured using a digital caliper (P01 110–120; ASONE, Osaka, Japan). Eggshell thickness (ST) was measured by a Peacock dial pipe gauge P-1 (Ozaki MFG Co., Ltd., Tokyo, Japan). Eggshell color lightness (SCL), redness (SCR), and yellowness (SCY) were measured using a chromameter (CR-10 Plus Color Reader; Konica Minolta Japan Inc., Tokyo, Japan). After measuring the egg traits, egg albumen and yolk samples were separately collected and frozen within 2–3 days after laying. Each sample was diluted 2-fold with distilled water (DW) and mixed. Each solution was kept in a tube at −30 °C until use.

### 2.6. Free Amino Acid Analysis of Egg Albumen

A total of 38 eggs (n = 20 at 7 wks and n = 18 at 14 wks) were used to analyze free amino acids in albumen. The albumen solution (250 μL) was mixed with 250 μL 16% trichloroacetic acid solution (FUJIFILM Wako Chemicals, Osaka, Japan). After mixing, it was centrifuged at 11,000× *g* for 15 min (model 2410; KUBOTA Corporation Co., Ltd., Osaka, Japan and RX series and RX II series; HITACHI Ltd., Tokyo, Japan). The supernatant was filtered out using a disposable cellulose acetate membrane filter unit with a 0.45 μm pore size (DISMIC-25CS; Advantec Toyo Kaisha, Ltd., Tokyo, Japan). The filtered supernatant (40 μL) was heated at 40 °C for 90 min in a vacuum oven (VOS-201SD, Eyela, Tokyo, Japan), and a 20 µL mixing solution (ethanol/DW/triethylamine = 2:2:1) was added. Then, the sample was vortexed for 20 min using a Micro Mixer E-36 (TAITEC Corporation, Saitama, Japan). After heating at 40 °C for 40 min in the vacuum oven again, 20 µL mixing solution (ethanol/DW/trimethylamine/phenylisothiocyanate = 7:1:1:1) was added. The sample was vortexed for 20 min again, and the sample was re-heated at 40 °C for 40 min in the vacuum oven. After preprocessing, the sample tube was placed at −30 °C until the sample was analyzed.

Free amino acids were analyzed by HPLC (LC-2010CHT; Shimadzu Co. Ltd., Kyoto, Japan). Two connecting columns, 25 cm column and 15 cm column, were used (TSKgel ODS80Ts, Tosoh Corporation, Tokyo, Japan). The detected value was set to 254 nm, and the column temperature was set to 40 °C. Two types of buffer, mobile phase A (60 mM acetate acid buffer/acetonitrile = 94:6, pH 5.60) and mobile phase B (60 mM acetate acid buffer/acetonitrile = 40:60, pH 6.95), were used. Amino acid standards (types H and B), including L-asparagine and L-glutamine (FUJIFILM Wako Chemicals, Osaka, Japan), were preprocessed using the same method of albumen. The standards were analyzed before every 25 samples, and the absolute concentrations of amino acids were calculated from the peaks of the sample and standard [26].

### 2.7. Free Amino Acid Analysis of Egg Yolk

A total of 39 eggs (n = 20 at 7 wks and n = 19 at 14 wks) were used to analyze free amino acids in the yolk. The yolk solution (5 mL) was mixed with 5 mL 16% trichloroacetic acid solution and then vortexed. The samples were centrifuged at 1400× *g* for 15 min. After centrifuging, the sample preprocessing and amino acid analysis was performed with the same method described above.

### 2.8. Fecal Corticosterone Extraction and Measurement

All feces obtained within the 2 h sampling time was placed in a Petri dish and dried at 60 °C for 2 days. Then, the samples were stored frozen until the extraction operation was performed. To extract corticosterone (CORT) in feces, dried feces were crushed and weighed at portions of 40 mg each in a 2.0 mL tube, and then were dispensed 1 mL of methanol. After that, it was stirred at room temperature for 24 h using a rotator. The stirred sample was centrifuged at 10,000× *g* at room temperature for 10 min. Then, the supernatant was transferred to a test tube (13 × 100 mm) and dried with nitrogen gas for about 40 min while heating at 38 °C in a heat block (Dry Thermo Unit1C, TAITEC, Koshigaya, Japan). PBS (0.3 mL) was added, and the mixture was shaken at room temperature for 1 h using the shaker. After shaking, it was transferred to a microtube and stored at −30 °C until measurement [30,31]. The concentration of corticosterone in the extracted feces was measured using an ELISA measurement kit (Corticosterone Enzyme Immunoassay Kit, ARBOR ASSAYS, Ann Arbor, MI, USA). Visible light density was measured at a wavelength of 450 nm using a microplate reader (I Mark microplate reader, Bio-Radola Volunteers Co., Ltd., Tokyo, Japan).

### 2.9. Statistical Analysis

The data (body weight gain, egg traits, albumen free amino acids, yolk free amino acids, and fecal corticosterone) were analyzed to test the housing effect using a one-way analysis of variance (ANOVA) with the R Studio. Data were shown as the mean ± SD. All data were analyzed separately for the first and second stages. Egg production rates were tested by one-way ANOVA in each stage. All statements of significance were based on a probability value equal to or less than 0.05. Finally, Pearson’s correlation analysis was performed with the data of BWG, egg traits, amino acid contents, and CORT obtained from the cage system using the “corrplot” package of R (*p* < 0.05). For the litter floor system, phenotypic correlations among BWG, egg traits, and amino acid contents were analyzed.

## 3. Results

### 3.1. Body Weight Gain

The effect of the housing systems (cage and litter) was tested on BWG in the first and second stages (Table 1). In the first stage, the BWG in the cage and litter systems were 130 ± 50 g and 174 ± 46 g, respectively. No significant difference was observed in the first stage. On the other hand, a significant difference was observed in the second stage (*p* < 0.001). The BWG of the cage system was 84 ± 52 g, whereas that of the litter system was −10 ± 41 g in the second stage.

### 3.2. Egg Production Rate

Egg production rates (%) in the cage and litter systems were plotted in Appendix A. At 0 wks, hens were 15 and 17 weeks of age (n = 5 each in the cage and litter systems). Therefore, hens started to lay from around 17 and 19 weeks of age (2 wks) in the cage system. In the litter system, hens started to lay from around 19 and 21 weeks of age (4 wks). There were no significant differences between the cage and litter systems in each stage (0–19 wks) of egg production rates, except for 3 wks (*p* < 0.05).

### 3.3. Egg Traits

To determine the effects of housing systems on egg traits, eggs from the cage and litter systems were analyzed (Table 1). In the first stage, there were significant differences in SW (3.7 ± 0.5 g and 3.3 ± 0.5 g in cage and litter hens, respectively) and SCL (73.4 ± 3.0 and 75.6 ± 2.8 in cage and litter hens, respectively) (*p* < 0.05). In the second stage, YW, ST, and SCL were significantly different (*p* < 0.05). The cage YW was heavier than the litter (9.0 ± 0.8 g and 8.5 ± 0.8 g in cage and litter hens, respectively). Although ST in the cage was thicker (0.41 ± 0.03 mm and 0.38 ± 0.03 mm in cage and litter, respectively), SCL in the litter was higher (75.9 ± 2.9 and 78.3 ± 2.5 in cage and litter, respectively). There was no difference between cage and litter hens in the remaining traits.

As advancing hens age, the sizes and weights of the egg tended to be larger in EW, LLE, LSE, YW, SW, and AW (Table 1). Eggshell colors in the second stage tended to be lighter (in SCL) and paler (in SCR and SCY) than those in the first stage. Even though the common aging effects tended to be seen, a robust effect of the housing system was found in eggshell color lightness (SCL). It is noteworthy that the litter conditions at 7 weeks made the eggshell color of those hens lighter than that of the cage Tosa-jidori hens.

Additionally, the index traits were shown in Appendix A.

### 3.4. Free Amino Acid Traits in Albumen

All 20 free amino acids were detected in egg albumen: aspartic acid (A_Asp), glutamic acid (A_Glu), asparagine (A_Asn), serine (A_Ser), glutamine (A_Gln), glycine (A_Gly), histidine (A_His), arginine (A_Arg), threonine (A_Thr), alanine (A_Ala), proline (A_Pro), GABA (A_GABA), tyrosine (A_Tyr), valine (A_Val), methionine (A_Met), cysteine (A_Cys), isoleucine (A_Ile), leucine (A_Leu), phenylalanine (A_Phe), and lysine (A_Lys) (Table 2). There were significant differences in free amino acid contents in the first stage among five amino acids, which are A_Gln, A_His, A_Met, A_Cys, and A_Lys (*p* < 0.05). The contents of five albumen amino acids in caged eggs were significantly higher than those in the litter.

In the second stage, there were significant differences between the cage and litter systems in seven albumen amino acids, which are A_Asp, A_Glu, A_Ser, A_Thr, A_Ala, A_Pro, and A_Phe (*p* < 0.05). The contents of seven albumen amino acids in the cage system were significantly higher than in the litter.

### 3.5. Free Amino Acid Traits in Yolk

All 20 free amino acids were detected in egg yolk: aspartic acid (Y_Asp), glutamic acid (Y_Glu), asparagine (Y_Asn), serine (Y_Ser), glutamine (Y_Gln), glycine (Y_Gly), histidine (Y_His), arginine (Y_Arg), threonine (Y_Thr), alanine (Y_Ala), proline (Y_Pro), GABA (Y_GABA), tyrosine (Y_Tyr), valine (Y_Val), methionine (Y_Met), cysteine (Y_Cys), isoleucine (Y_Ile), leucine (Y_Leu), phenylalanine (Y_Phe), and lysine (Y_Lys) (Table 3). No significant effect of the housing systems was observed on the yolk free amino acid contents in both the first and second stages (*p* > 0.05).

### 3.6. Fecal Corticosterone

The effect of the housing systems was tested on the amount of fecal CORT in the cage and litter systems (Figure 2). There was no significant difference (F_1,18_ = 0.003, *p* = 0.955) between the cage (56.1 ± 20.5 pg/mg) and litter (56.6 ± 15.0 pg/mg) in fecal corticosterone.

### 3.7. Phenotypic Correlation Analysis

Phenotypic correlations among 10 egg traits, 20 yolk amino acids, 20 albumen amino acids, BWG, and CORT were estimated using cage hens in the second stage (Figure 3). There were positive correlations among the sizes and weights of the egg. Positive correlations were observed among most yolk amino acid traits. On the other hand, there were just a few correlations among albumen amino acid traits. There were no significant correlations between fecal corticosterone and the others.

Correlations among 10 egg traits, 20 yolk amino acids, 20 albumen amino acids, and body weight gain were estimated using cage hens in the first stage (Appendix A), whereas correlations among 10 egg traits, 20 yolk amino acids, and 20 albumen amino acids were estimated using litter floor hens in the first and second stages (Appendix A, respectively). A similar tendency with Figure 3 was seen in Appendix A. In particular, positive phenotypic correlations among most albumen amino acid traits were seen in litter floor hens in the first stage (Appendix A).

## 4. Discussion

We evaluated the effects of housing systems on BWG, 10 egg traits, 20 kinds of albumen and yolk amino acid contents, and CORT in Tosa-jidori by switching the housing systems (cage and litter) between the first and second stages. BWG, eggshell weight, yolk weight, eggshell thickness, eggshell color lightness, and twelve amino acid traits (A_Gln, A_His, A_Met, A_Cys, A_Lys, A_Asp, A_Glu, A_Ser, A_Thr, A_Ala, A_Pro, and A_Phe) significantly differed between cage and litter hens. These results suggest that eggshell lightness and several egg traits, including albumen amino acid contents, can be changed by using the different housing systems.

This study revealed no difference in BWG in the first stage but a significant difference in the second stage. Cage hens had higher BWG than litter hens. In this study, the average BW ± SD of Tosa-jidori hens at 14 wks (29–31 wks of age) was 804 ± 59 g and 714 ± 82 g in cage and litter, respectively. We have reported the BW of Tosa-jidori hens (n = 17) at 32 wks of age was 679 ± 110 g in the cage system [35]. Therefore, the present Tosa-jidori hens at 14 wks (29–31 wks of age) in the cage system would be of a relatively higher BW than usual in our animal facility. Several reports compare BW between the cage and non-cage systems. Singh et al. [36] reported production performance and egg quality of four strains of laying hens, including Lohmann White, H&N White, and Lohmann Brown, in conventional cages and floor pens, which indicates that layer hens in floor pens had significantly higher BW than conventional cage hens throughout the experimental stages (at 20–50 weeks of age). Yilmaz Dikmen et al. [6] reported that hens in the free range had significantly higher final BW (at 66 weeks of age) than Lohmann Brown laying hens in a conventional cage, although there was no difference in the initial BW (at 17 weeks of age). These previous results in BW were different from our present results. One of the differences is breed/strain, since the previous and present studies used layer strains and an indigenous breed, respectively. Moreover, the previous studies were carried out for 30–50 wks, whereas the present study was conducted for 14 wks. Therefore, the differences in genetic background and/or experimental periods may lead to discrepancies between the previous and present studies. In the future, we should carry out another long-term experiment to understand the relationship between body weight gain and the housing system.

A robust effect of switching the housing systems on SCL was found in this study. Similar results of housing systems have been reported [37]. Sekeroglu et al. [37] reported that SCL in deep litter and the free range was significantly lighter than in the cage system using brown laying hens. Eggshell color is known as a heritable trait [38], and several quantitative trait loci (QTLs) for eggshell color have been reported [39]. Eggshell color is also affected by many environmental factors, such as housing systems, nutrition, and environmental pollutants [40]. The principal pigments of eggshells are biliverdin and protoporphyrin [41]. Protoporphyrin has been negatively correlated with eggshell lightness [40], which means the change in eggshell lightness in this study may be involved with the concentration of protoporphyrin accumulated. In the future, investigation of not only eggshell color but also protoporphyrin metabolism may help to clarify the causes of increasing eggshell lightness by litter housing systems.

In the other egg traits, SW significantly differed between cage and litter in the first stage, whereas YW and ST significantly differed in the second stage. The trait values of SW and ST in caged eggs were significantly higher than in litter eggs. The same feed materials were used in both cage and litter in this study; nevertheless, the eggshell quality was changed. It is reported that correct calcium requirements in the diets of hens may be changed by different housing systems [42]. Therefore, the calcium intake and/or metabolism in litter hens might not be enough to create more eggshells.

This study revealed that free amino acids in albumen were changed by the housing systems. We previously confirmed the significant effects of genetic (breed) and environmental (feed) factors on yolk and albumen amino acid traits in several research environments [21,22,23,24,25,26]. This study newly revealed the possibility that the housing system itself affects albumen amino acid contents as one of the environmental factors, alongside our previous study [43]. Cage eggs contained higher albumen amino acids (i.e., A_Gln, A_His, A_Met, A_Cys, and A_Lys in the first stages and A_Asp, A_Glu, A_Ser, A_Thr, A_Ala, A_Pro, and A_Phe in the second stage) than litter eggs. Although no amino acid was commonly found in both stages, the results imply that housing systems have the potential to change amino acid metabolites in albumen. Glu and Asp are known to relate to umami and sour tastes, whereas several amino acids are related to sweet (Ala, Ser, Thr, Pro, and Met) and bitter tastes (Phe, Lys, and His) [44,45]. Although it is not clear how the difference in albumen in this study tastes according to human perception, taste change may occur in different housing systems. Various reasons can be thought of when considering the causes of changes in egg components depending on the housing systems. For example, it is known that intestinal bacterial flora was modulated by environmental conditions such as different exercise stimuli [46] and that different housing systems change the microbiota in chickens [47,48]. Wan et al. [48] have reported differences in the cecal and duodenal microbiota compositions of Shendan chickens reared in different non-cage housing systems (plastic net housing system and floor litter housing system). This may indicate that different housing systems change the microbiota, which affects digestibility, nutrient absorption, metabolism, and egg formation. In the future, it will be necessary to analyze not only the egg components (final livestock product) but also the differences in the intestinal flora and metabolism to understand the mechanism of the differences.

Fecal corticosterone has been analyzed for stress evaluation in animals. Blood corticosterone concentration has been widely used in physiological stress evaluation [49]. In recent years, the metabolites (saliva, urine, feces, milk, and hair) of domestic animals have been used [30] because corticosterone measurements using blood samples represent concentrations at only a single point in time [50]. In addition, a highly invasive blood sampling operation itself causes acute stress, which increases blood corticosterone [29]. This study suggests that there may be no difference in stress levels depending on the housing systems shown by corticosterone in feces. On the other hand, Alm et al. [51] have reported that the addition of 3% crumbled straw pellets in the feed increased fecal corticosterone metabolite concentration in white and brown layer hybrids, which indicates that small changes in diet and fiber content can influence fecal corticosterone levels. In this study, rice husks were used as litter materials on the floor. Although it is unclear whether rice husks certainly were eaten by litter hens, they would contribute to the increase in corticosterone in feces in the litter system only. In the future, it will be necessary to further evaluate the fecal corticosterone and/or egg albumen corticosterone [29] using mesh floor barn and cage systems without any fiber content.

In the phenotypic correlation analyses, there were moderate to high positive correlations among the traits of yolk amino acids in both housing systems in this study. These results were similar to our previous reports using a variety of breeds [23,24,26]. On the other hand, phenotypic correlations among albumen amino acid traits were slightly different from the previous studies. Although we have reported moderate to high positive correlations among albumen amino acid traits as well as the yolk amino acids [23,24,26], a similar tendency was seen in litter hens in the first stage only (Appendix A). In the remaining three cases (Figure 3 and Appendix A), there were just a few correlations among albumen amino acid traits, which were slightly different from our previous results. Since Tosa-jidori hens were used in this study, different results might be seen in comparison with the other breeds of chickens. In addition, we cannot find any egg traits that were correlated with fecal corticosterone levels in this study. Positive correlations (r = 0.87) between blood plasma corticosterone and egg albumen corticosterone have been reported [29]. If we find some correlated egg indicators with stress levels, non-invasive stress evaluation will be available by using daily collected egg samples in litter and cage hens.

## 5. Conclusions

In conclusion, this study indicated that housing systems affect eggshell lightness and several egg traits, including albumen amino acid traits using the Tosa-jidori model of chickens. Since it is a period of change for housing systems around the world, the results of this study will be important knowledge in the future of poultry science. Although the present designer egg production tends to focus on feeding materials in layers, this study indicates the possibility of selecting another option to change egg albumen amino acid contents by using the housing system. Further efforts in investigating the effects of the housing system, including microbiome in both the commercial layer industry and several genetically different breeds of chickens, will give us valuable insights for future egg production.

## Figures and Tables

**Figure 1 animals-13-01837-f001:**
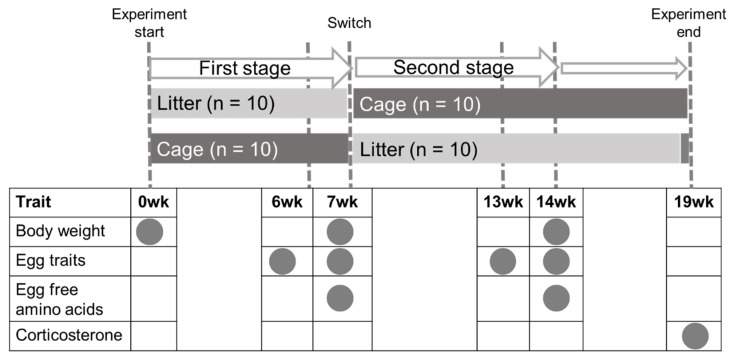
Experimental design. Two production stages are defined as the first stage (0 to 7 wks) and the second stage (7 to 14 wks). Each stage lasts for 7 weeks until the egg sampling. Tosa-jidori hens are divided into two housing system groups, cage (n = 10) and litter (n = 10) at 0 wks. After collecting eggs at 7 wks, the housing systems are switched to start the second stage. Body weight is measured at 0, 7, and 14 wks. Eggs are collected at 6, 7, 13, and 14 wks to measure egg traits. Egg-free amino acids are measured using eggs at 7 and 14 wks. Corticosterone is measured using a fecal sample collected at 19 wks.

**Figure 2 animals-13-01837-f002:**
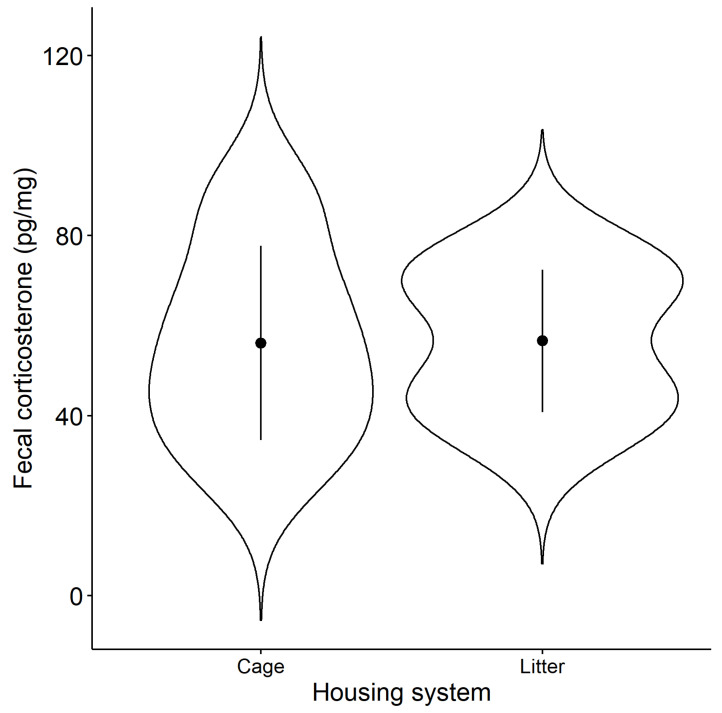
Fecal corticosterone concentration in different housing systems. A violin plot is used to show the data distribution of corticosterone concentration in feces. Black dots and bars indicate the mean ± SD for each housing system. There is no significant effect on housing systems.

**Figure 3 animals-13-01837-f003:**
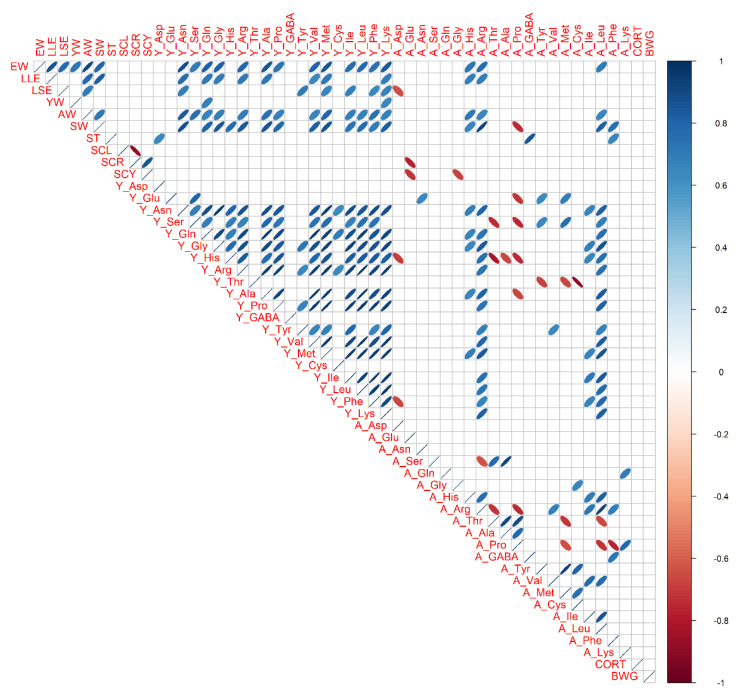
Phenotypic correlations of cage hens in the second stage. Egg traits, yolk amino acids, albumen amino acids, corticosterone, and body weight gain from battery cage hens in the second stage are used. Trait abbreviations are indicated in the main text. Pearson’s correlations are expressed by ellipses. Blue and red ellipses indicate positive and negative correlations in each pair, respectively (*p* < 0.05). Blank means no correlations.

**Table 1 animals-13-01837-t001:** Body weight gain and egg traits measured in first and second stages of two housing systems.

Traits	Cage	Litter	One-Way ANOVA				
			df_bet_	df_res_	F Value	*p* Value	
First stage	(n = 20)	(n = 20)					
Body weight gain (g)	130 ± 50	174 ± 46	1	18	3.801	0.067	
Egg weight (g)	25.5 ± 2.7	24.7 ± 1.3	1	38	1.173	0.286	
Length of long axis of the egg (mm)	42.7 ± 1.9	42.2 ± 1.5	1	38	1.065	0.309	
Length of short axis of the egg (mm)	32.3 ± 1.1	32.4 ± 0.7	1	38	0.115	0.736	
Yolk weight (g)	7.5 ± 0.8	7.2 ± 0.5	1	38	1.708	0.199	
Eggshell weight (g)	3.7 ± 0.5	3.3 ± 0.5	1	38	5.080	0.030	*
Albumen weight (g)	14.3 ± 1.8	14.2 ± 1.3	1	38	0.112	0.739	
Eggshell thickness (mm)	0.38 ± 0.04	0.37 ± 0.03	1	38	0.248	0.621	
Eggshell color lightness	73.4 ± 3.0	75.6 ± 2.8	1	38	5.646	0.023	*
Eggshell color redness	8.0 ± 2.0	7.7 ± 2.3	1	38	0.193	0.663	
Eggshell color yellowness	18.0 ± 2.5	17.7 ± 2.7	1	38	0.123	0.727	
Second stage	(n = 20)	(n = 19)					
Body weight gain (g)	84 ± 52	−10 ± 41	1	18	18.06	<0.001	*
Egg weight (g)	29.2 ± 1.9	28.6 ± 2.8	1	37	0.559	0.459	
Length of long axis of the egg (mm)	45.0 ± 1.6	43.9 ± 1.5	1	37	3.860	0.057	
Length of short axis of the egg (mm)	34.2 ± 0.8	34.1 ± 1.0	1	37	0.102	0.751	
Yolk weight (g)	9.0 ± 0.8	8.5 ± 0.8	1	37	4.230	0.047	*
Eggshell weight (g)	4.0 ± 0.4	3.9 ± 0.4	1	37	1.391	0.246	
Albumen weight (g)	16.2 ± 1.3	16.3 ± 1.8	1	37	0.052	0.820	
Eggshell thickness (mm)	0.41 ± 0.03	0.38 ± 0.03	1	37	4.559	0.039	*
Eggshell color lightness	75.9 ± 2.9	78.3 ± 2.5	1	37	7.126	0.011	*
Eggshell color redness	6.3 ± 1.8	5.5 ± 1.8	1	37	1.681	0.203	
Eggshell color yellowness	15.5 ± 2.4	14.5 ± 2.8	1	37	1.214	0.278	

* *p* < 0.05. Mean ± SD. df_bet_: between groups degree of freedom. df_res_: residual degree of freedom.

**Table 2 animals-13-01837-t002:** Albumen amino acids measured in first and second stages in two housing systems.

Amino Acid	Cage	Litter	One-Way ANOVA				
(μg/mL)			df_bet_	df_res_	F Value	*p* Value	
First stage	(n = 10)	(n = 10)					
A_Asp	2.0 ± 0.4	2.6 ± 2.2	1	18	0.542	0.471	
A_Glu	3.3 ± 0.4	4.5 ± 3.6	1	18	1.070	0.315	
A_Asn	0.5 ± 0.1	0.5 ± 0.1	1	18	0.514	0.482	
A_Ser	1.4 ± 0.7	1.6 ± 1.4	1	18	0.176	0.679	
A_Gln	1.2 ± 0.2	0.6 ± 0.2	1	18	53.100	<0.001	*
A_Gly	0.7 ± 0.2	0.5 ± 0.3	1	18	1.385	0.255	
A_His	2.4 ± 0.1	1.5 ± 0.4	1	18	32.300	<0.001	*
A_Arg	2.3 ± 0.3	2.3 ± 1.3	1	18	0.008	0.931	
A_Thr	0.3 ± 0.2	0.7 ± 0.9	1	18	1.889	0.186	
A_Ala	0.7 ± 0.3	1.0 ± 0.9	1	18	1.005	0.329	
A_Pro	0.6 ± 0.1	1.6 ± 1.4	1	18	4.124	0.057	
A_GABA	0.4 ± 0.5	0.5 ± 0.3	1	18	0.034	0.857	
A_Tyr	5.7 ± 1.4	6.7 ± 1.2	1	18	2.422	0.137	
A_Val	3.4 ± 0.7	3.0 ± 1.8	1	18	0.368	0.552	
A_Met	9.4 ± 1.2	6.6 ± 1.2	1	18	24.760	<0.001	*
A_Cys	10.2 ± 0.3	6.7 ± 0.4	1	18	421.500	<0.001	*
A_Ile	1.9 ± 0.4	2.6 ± 1.6	1	18	2.073	0.167	
A_Leu	6.1 ± 1.3	6.9 ± 3.6	1	18	0.371	0.550	
A_Phe	7.7 ± 2.3	8.2 ± 3.7	1	18	0.118	0.735	
A_Lys	3.0 ± 0.1	1.6 ± 0.3	1	18	176.600	<0.001	*
Second stage	(n = 10)	(n = 8)					
A_Asp	1.3 ± 0.4	0.8 ± 0.1	1	16	13.320	0.002	*
A_Glu	3.0 ± 0.3	2.5 ± 0.2	1	16	15.110	0.001	*
A_Asn	0.5 ± 0.1	0.6 ± 0.1	1	16	2.484	0.135	
A_Ser	1.2 ± 0.2	0.9 ± 0.1	1	16	20.650	<0.001	*
A_Gln	1.0 ± 0.1	1.0 ± 0.1	1	16	0.032	0.860	
A_Gly	0.6 ± 0.1	0.6 ± 0.1	1	16	0.001	0.975	
A_His	1.7 ± 0.2	1.7 ± 0.1	1	16	0.188	0.670	
A_Arg	3.2 ± 0.4	3.1 ± 0.3	1	16	1.061	0.318	
A_Thr	0.7 ± 0.2	0.6 ± 0.1	1	16	6.172	0.024	*
A_Ala	0.8 ± 0.1	0.6 ± 0.0	1	16	15.380	0.001	*
A_Pro	0.9 ± 0.2	0.7 ± 0.1	1	16	6.114	0.025	*
A_GABA	0.5 ± 0.7	0.1 ± 0.2	1	16	2.538	0.131	
A_Tyr	8.4 ± 3.5	10.0 ± 2.3	1	16	1.059	0.319	
A_Val	2.0 ± 0.2	1.8 ± 0.2	1	16	4.474	0.050	
A_Met	11.7 ± 1.1	12.0 ± 0.5	1	16	0.352	0.561	
A_Cys	7.1 ± 1.0	7.8 ± 0.5	1	16	3.837	0.068	
A_Ile	3.0 ± 0.5	3.0 ± 0.4	1	16	0.020	0.888	
A_Leu	7.4 ± 1.5	6.4 ± 1.4	1	16	1.723	0.208	
A_Phe	8.1 ± 1.7	5.5 ± 1.4	1	16	10.030	0.006	*
A_Lys	1.4 ± 0.1	1.4 ± 0.1	1	16	0.001	0.974	

* *p* < 0.05. Mean ± SD. df_bet_: between groups degree of freedom. df_res_: residual degree of freedom.

**Table 3 animals-13-01837-t003:** Yolk amino acids measured in first and second stages of two housing systems.

Amino Acid	Cage	Litter	One-Way ANOVA			
(μg/mL)			df_bet_	df_res_	F Value	*p* Value
First stage	(n = 10)	(n = 10)				
Y_Asp	52.4±6.3	57.7 ± 8.9	1	18	2.096	0.165
Y_Glu	206.6 ± 7.7	214.8 ± 21.4	1	18	1.184	0.291
Y_Asn	37.2 ± 1.8	35.2 ± 3.9	1	18	1.823	0.194
Y_Ser	76.6 ± 3.9	80.7 ± 7.9	1	18	2.034	0.171
Y_Gln	68.9 ± 2.9	65.3 ± 7.2	1	18	1.934	0.181
Y_Gly	23.9 ± 1.4	24.2 ± 2.4	1	18	0.166	0.688
Y_His	22.3 ± 2.1	23.8 ± 1.9	1	18	2.585	0.125
Y_Arg	88.8 ± 7.8	91.1 ± 10.7	1	18	0.251	0.622
Y_Thr	69.8 ± 10.3	57.0 ± 9.3	1	11	4.178	0.066
Y_Ala	49.6 ± 2.6	50.0 ± 4.3	1	18	0.041	0.842
Y_Pro	44.7 ± 3.2	47.3 ± 4.9	1	18	1.703	0.208
Y_GABA	8.0 ± 1.6	8.2 ± 1.2	1	18	0.098	0.758
Y_Tyr	79.0 ± 4.6	80.3 ± 8.4	1	18	0.144	0.709
Y_Val	75.9 ± 9.1	74.6 ± 12.6	1	18	0.058	0.812
Y_Met	30.1 ± 3.4	29.0 ± 3.9	1	18	0.453	0.510
Y_Cys	5.4 ± 0.4	5.4 ± 0.2	1	18	0.020	0.890
Y_Ile	60.1 ± 6.9	59.8 ± 7.9	1	18	0.006	0.938
Y_Leu	121.2 ± 11.9	118.7 ± 12.9	1	18	0.179	0.678
Y_Phe	47.4 ± 5.3	46.9 ± 6.5	1	18	0.032	0.860
Y_Lys	115.8 ± 12.8	115.6 ± 18.1	1	18	0.001	0.976
Second stage	(n = 10)	(n = 9)				
Y_Asp	63.0 ± 7.9	58.4 ± 11.4	1	17	0.942	0.345
Y_Glu	245.0 ± 15.8	229.4 ± 28.0	1	17	2.048	0.171
Y_Asn	48.0 ± 3.3	47.4 ± 5.8	1	17	0.058	0.813
Y_Ser	98.4 ± 7.8	98.7 ± 11.4	1	17	0.005	0.943
Y_Gln	83.1 ± 4.8	79.2 ± 10.0	1	17	1.049	0.320
Y_Gly	29.3 ± 2.1	28.5 ± 3.5	1	17	0.335	0.571
Y_His	33.9 ± 3.1	34.2 ± 4.7	1	17	0.032	0.861
Y_Arg	122.8 ± 11.8	116.4 ± 19.6	1	17	0.687	0.419
Y_Thr	91.8 ± 13.0	87.3 ± 18.1	1	17	0.342	0.566
Y_Ala	67.1 ± 6.0	62.2 ± 9.0	1	17	1.738	0.205
Y_Pro	45.5 ± 3.3	45.6 ± 5.8	1	17	0.002	0.968
Y_GABA	1.4 ± 0.7	1.0 ± 0.1	1	17	2.519	0.131
Y_Tyr	91.4 ± 5.8	89.6 ± 11.9	1	17	0.170	0.685
Y_Val	111.1 ± 8.6	105.2 ± 16.9	1	17	0.860	0.367
Y_Met	41.5 ± 4.0	40.4 ± 6.6	1	17	0.153	0.701
Y_Cys	3.4 ± 0.2	3.2 ± 0.1	1	17	2.620	0.124
Y_Ile	75.3 ± 7.3	75.4 ± 11.4	1	17	0.000	0.985
Y_Leu	160.0 ± 15.3	157.2 ± 24.8	1	17	0.084	0.776
Y_Phe	64.6 ± 5.3	66.0 ± 9.9	1	17	0.146	0.708
Y_Lys	156.7 ± 15.8	155.9 ± 25.1	1	17	0.007	0.936

Mean ± SD. df_bet_: between groups degree of freedom. df_res_: residual degree of freedom.

## Data Availability

Data are available upon request to the authors.

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
