# Peer review of "Housing Systems Affect Eggshell Lightness and Free Amino Acid Contents of Egg Albumen in Tosa-Jidori Chickens: A Preliminary Research"

_animals, 2023, doi:10.3390/ani13111837_

Round 1

Reviewer 1 Report

INTRODUCTION:

It would be useful to include a brief description of Tosa-jidori hens.

The purpose of the relocation should be clearly written. Why was this needed? Was it possible to implement such an application in the sector? It should be explained.

Line 61: “reference 11”, There is no information on egg production in the mentioned reference. Refine the reference once again.

Line 63: add the reference “Japanese annual per capita consumption of hen’s eggs is the second largest after Mexico.”

MATERİAL AND METHODS:

Lines 97-100: It appears that there are not enough animals for this trial. Why did the experiment use so few animals? It is important to provide details about this genotype's traits. The daily lighting time is very high at 17 weeks.

Why are the alternatives to the conventional cage system, the enriched cage system and the free-range system, not included?

Line 94: “were reared in group housing cages with”. Conventional cages?

Lines 100-101: It is appropriate to provide the average body weight.

Lines 106-107: Replace “35*18 cm2/hen) with “630 cm2/hen”

Lines 108: Replace    “(450 × 250 cm2 /10 hens” with “11.25 m2/10 hens”

Lines 114-123: It would be more appropriate to specify the number of weeks in the method based on the hens' age. Because it is more appropriate to follow the characteristics of performance and egg quality according to the hens' ages.

Figure 1: According to the title, it is stated that two egg production systems are compared. However, in the material method, animals were switched between these systems. Such an application is not suitable for the sector. At 16 to 18 weeks of age, layer hens are moved to egg production housing, where they continue to lay eggs. There is an economic loss when you switch animals that have begun to lay eggs in the cage system to the deep litter system because the rate of floor eggs, dirty eggs, and broken eggs becomes too high.

Figure 1: Why was body weight determined at week 14 rather than at the end of the trial (19 weeks)?

Lines 145-155: How many days of eggs were collected and under what conditions they were stored should be specified for egg quality analysis.

Lines 145-155: Why didn't you determine egg albumen height, haugh unit, and breaking strength?

Table 1: Please give the body weight gain in grams.

Table 1: Egg weight is very low. Therefore, it is necessary to give information about some characteristics of the genotype.

Table 1: Instead of “Length of long axis of the egg”, the shape index must be provided.

Table 1: Please give the “yolk weight, eggshell weight, and albumen weight” in %.

Table 1: There are only two hypotheses in statistics, they are H0 and H1. In the null hypothesis (H0), the distribution is gaussian, the treatment effect is not significant. In the H1 hypothesis, the situation is the opposite. In the natural, health, and engineering sciences, the acceptance probability of the null hypothesis spreads up to 0.95/1. If a P value of 0.049 is found in a study with a significance level of 0.05, H0 is rejected. But this does not mean that the "more significant" difference occurs when a P value of 0.01 or 0.001 is found in the same study. There is no H2 or H3 hypotheses in statistics science. Therefore, use only one of these levels (0.05, 0.01, 0.001) in your studies.

Lines 250-254: Could you review this sentence again? Because, according to the table, no statistical analysis was made to compare the first and second sections, and no statistical results were given in the table.

CONCLUSIONS:

It should be rewritten according to the results of the study.

You said that “In conclusion, this study firstly indicated that housing systems affect eggshell light- 434 ness and several egg traits including albumen amino acid traits.” There are many studies on the effect of the housing system on egg production and quality.

You said “Since it is the change time for housing systems in the world, the results of this study will be important knowledge in the future layer industry.” The practice is not suitable for the poultry industry. In addition, working with local genotypes and a small number of materials is insufficient for generalization in terms of sectors.

Author Response

Housing systems affect eggshell lightness and free amino acid contents of egg albumen in Tosa-jidori chickens

Nonoka Kawamura 1, Masahiro Takaya 1,2, Hideaki Hayashi 3 and Tatsuhiko Goto 1,4,*

<Reviewer 1>

INTRODUCTION:

It would be useful to include a brief description of Tosa-jidori hens.

Response 1:

We added the information of Tosa-jidori (L86-94). Tosa-jidori is one of the smallest breed in Japanese indigenous chickens. Therefore, the egg is small.

The purpose of the relocation should be clearly written. Why was this needed? Was it possible to implement such an application in the sector? It should be explained.

Response 2:

Actually, relocation (switching the housing system) is not usual in the layer industry. We would like to collect experimental evidence using the Tosa-jidori model. We added the description as below (L92-98).

To investigate the effect of the housing system, we adopted switching the housing systems (cage and litter) using the Tosa-jidori model. The main reason for the switching experiment is to determine the effect of the housing systems experimentally. If the housing system has some strong effects on egg quality traits, the significant effects on egg traits can be seen in both the first and second stages (2 months each) after switching the housing systems.

Line 61: “reference 11”, There is no information on egg production in the mentioned reference. Refine the reference once again.

Response 3:

“Egg production quantity in national level” can be searched using FAOSTAT. After inputting the item name “Hen egg in shell, fresh” and we could see the egg data. We think this reference is suitable.

Line 63: add the reference “Japanese annual per capita consumption of hen’s eggs is the second largest after Mexico.”

Response 4:

We added the below reference.

Neves et al. (2021 International Food and Agribusiness Management Review: 24, 138-161. DOI: 10.22434/IFAMR2020.0031).

MATERİAL AND METHODS:

Lines 97-100: It appears that there are not enough animals for this trial. Why did the experiment use so few animals? It is important to provide details about this genotype's traits. The daily lighting time is very high at 17 weeks.

Response 5:

Actually, we agree with your suggestion to increase sample size.

In this study, a model study using the Tosa-jidori was conducted to test the effects of deep litter on egg traits. It was difficult to increase the sample size because egg samples were not collected in the industrial layer farm. Although the sample size is not large in this study, a model experiment by the switching housing environment will provide us a unique evidence (this cannot be obtained in the industrial setting).

Genotype’s characteristics are shown in Introduction (L86-92). The daily lighting time is set the same to the adult chickens in this facility (Ono et al., J Poult Sci 2022, 59, 38-47.).

Why are the alternatives to the conventional cage system, the enriched cage system and the free-range system, not included?

Response 6:

The animal facility in our University has a limited space. We managed only two housing systems (cage and deep litter) until now. If we added several evidence on housing system effect of egg traits, we will make additional space to test enriched cage in the future.

Line 94: “were reared in group housing cages with”. Conventional cages?

Response 7:

“Conventional cages”. We reworded (L107).

Lines 100-101: It is appropriate to provide the average body weight.

Response 8:

Although we mentioned only body weight gain, we added the average body weight as below.

Mean ± SD of the body weights were 594 ± 86.3 g and 546 ± 55.2 g in cage and litter, respectively (L114-115).

Lines 106-107: Replace “35*18 cm2/hen) with “630 cm2/hen”

Response 9:

We replaced (L120).

Lines 108: Replace    “(450 × 250 cm2 /10 hens” with “11.25 m2/10 hens”

Response 10:

We replaced (L122).

Lines 114-123: It would be more appropriate to specify the number of weeks in the method based on the hens' age. Because it is more appropriate to follow the characteristics of performance and egg quality according to the hens' ages.

Response 11:

Since we have both 17wk and 15wk hens (L110), we thought it would be complicated to indicate their age in weeks. Therefore, we wrote in weeks of the experiment.

However, it should be better to describe more information about the laying stage. So, we added the sentences as below.

(L128-130): The timing of starting the experiment is set at 0 wk, which is the early laying stage (n = 5 at 17 wk and n = 5 at 15 wk in each group, as mentioned above).

(L139-140): The timing of ending the experiment is the middle laying stage (n = 5 at 36 wk and n = 5 at 34 wk in each group).

Figure 1: According to the title, it is stated that two egg production systems are compared. However, in the material method, animals were switched between these systems. Such an application is not suitable for the sector. At 16 to 18 weeks of age, layer hens are moved to egg production housing, where they continue to lay eggs. There is an economic loss when you switch animals that have begun to lay eggs in the cage system to the deep litter system because the rate of floor eggs, dirty eggs, and broken eggs becomes too high.

Response 12:

The aim of this study is to collect robust evidence how the deep litter affects egg traits under the experimental setting. Because of this purpose, the switching the housing system will provide us good knowledge using the Tosa-jidori model. We suppose 2 months experience will be enough time to change egg quality regardless of the prior experience.  

Figure 1: Why was body weight determined at week 14 rather than at the end of the trial (19 weeks)?

Response 13:

To compare performance at first and second stages, weight measurements including body weight, egg weight, yolk and albumen weights, were taken at the same week.

Lines 145-155: How many days of eggs were collected and under what conditions they were stored should be specified for egg quality analysis.

Response 14:

All eggs were collected daily. The eggs were stored on room temperature and then were measured egg traits. Albumen and yolk samples were frozen within 2-3 days. All the samples from both cage and litter were frozen under the same schedule.

Lines 145-155: Why didn't you determine egg albumen height, haugh unit, and breaking strength?

Response 15:

Unfortunately, we do not have equipment for measuring albumen height, haugh unit, and eggshell strength. We will buy the equipment in near future.

Table 1: Please give the body weight gain in grams.

Response 16:

Body weight gain was changed from kg to g (L239-242; Table 1).

Table 1: Egg weight is very low. Therefore, it is necessary to give information about some characteristics of the genotype.

Response 17:

According to your advices, we added the basic information about Tosa-jidori (L86-94).

Table 1: Instead of “Length of long axis of the egg”, the shape index must be provided.

Response 18:

Egg shape index (Mean ± SD) can be calculated.

(Cage in the first stage: 75.7 ± 2.1; Litter in the first stage: 76.9 ± 2.6; Cage in the second stage: 76.0 ± 2.4; Litter in the second stage: 77.5 ± 2.1)

Table 1: Please give the “yolk weight, eggshell weight, and albumen weight” in %.

Response 19:

Mean ± SD in YW%, SW%, and AW% can be calculated.

YW%: Cage in the first stage: 29.5 ± 1.5; Litter in the first stage: 29.3 ± 2.3; Cage in the second stage: 30.8 ± 2.0; Litter in the second stage: 29.6 ± 1.2

SW%: Cage in the first stage: 14.4 ± 1.6; Litter in the first stage: 13.3 ± 1.3; Cage in the second stage: 13.8 ± 0.9; Litter in the second stage: 13.5 ± 0.9

AW%: Cage in the first stage: 56.1 ± 2.4; Litter in the first stage: 57.3 ± 1.9; Cage in the second stage: 55.4 ± 2.2; Litter in the second stage: 56.8 ± 1.4

Table 1: There are only two hypotheses in statistics, they are H0 and H1. In the null hypothesis (H0), the distribution is gaussian, the treatment effect is not significant. In the H1 hypothesis, the situation is the opposite. In the natural, health, and engineering sciences, the acceptance probability of the null hypothesis spreads up to 0.95/1. If a P value of 0.049 is found in a study with a significance level of 0.05, H0 is rejected. But this does not mean that the "more significant" difference occurs when a P value of 0.01 or 0.001 is found in the same study. There is no H2 or H3 hypotheses in statistics science. Therefore, use only one of these levels (0.05, 0.01, 0.001) in your studies.

Response 20:

We revised P < 0.05 only according your suggestion.

Lines 250-254: Could you review this sentence again? Because, according to the table, no statistical analysis was made to compare the first and second sections, and no statistical results were given in the table.

Response 21:

Yes. Since we adopted the switching housing system in this study, the same individuals were used in cage at first stage and litter at second stage vice versa. Therefore, we did not conduct statistical analysis between the first and second sections to focus on the effect of the housing system at each stage. Because of no statistical analysis between the stages, we used the term “tended to” and did not use “significant”.

In general, egg weight and egg size increase along with advancing hens’ age.

Even in the ageing effect, the robust results were seen in eggshell color lightness. Because we would like to stress the robust effect of eggshell lightness, we mentioned the tendency (without statistical analysis) in this paragraph.

CONCLUSIONS:

It should be rewritten according to the results of the study. You said that “In conclusion, this study firstly indicated that housing systems affect eggshell light-ness and several egg traits including albumen amino acid traits.” There are many studies on the effect of the housing system on egg production and quality.

Response 22:

Regarding the results of albumen amino acids, it looks the first evidence. However, there are many evidence that housing system affect egg quality. Therefore, we deleted “firstly”.

You said “Since it is the change time for housing systems in the world, the results of this study will be important knowledge in the future layer industry.” The practice is not suitable for the poultry industry. In addition, working with local genotypes and a small number of materials is insufficient for generalization in terms of sectors.

Response 23:

The present model study using Tosa-jidori under the switching housing system is actually far from the Brown layer in the layer industry. However, we think that the experiment using switching housing system will provide important knowledge in the future “poultry science” in animal science field. Therefore, we reworded (L449).

Submission Date

09 April 2023

Date of this review

12 Apr 2023 13:20:33

Reviewer 2 Report

This manuscript presents a study that aimed to investigate the effects of housing systems (conventional cage vs. floor rearing) on egg quality traits and free amino acid contents of albumen and yolk. It is very interesting subject since there is not much information related to this issue as the authors also mentioned in the introduction. The manuscript is generally well-written and presents a clear research objective.

The use of Tosa-jidori hens as the experimental animals is appropriate for the study's aim, as it focuses on a specific breed. However, it is unfortunate that the number of experimental animals was small.

Overall, the manuscript  provides valuable insights into the effects of different housing systems on egg quality traits. The study's findings have implications for the egg industry and provide a foundation for future research in this area. 

Author Response

We appreciate it for your valuable comments to improve our manuscript. 

animals

Housing systems affect eggshell lightness and free amino acid contents of egg albumen in Tosa-jidori chickens

Nonoka Kawamura 1, Masahiro Takaya 1,2, Hideaki Hayashi 3 and Tatsuhiko Goto 1,4,*

<Reviewer 2>

This manuscript presents a study that aimed to investigate the effects of housing systems (conventional cage vs. floor rearing) on egg quality traits and free amino acid contents of albumen and yolk. It is very interesting subject since there is not much information related to this issue as the authors also mentioned in the introduction. The manuscript is generally well-written and presents a clear research objective.

Response 1:

We appreciate it for your contribution to our paper. Thank you very much to give us your time.

The use of Tosa-jidori hens as the experimental animals is appropriate for the study's aim, as it focuses on a specific breed. However, it is unfortunate that the number of experimental animals was small.

Response 2:

In this study, a model study using the Tosa-jidori was conducted to test the effects of deep litter on egg traits. It was difficult to increase the sample size because egg samples were not collected in the industrial layer farm. Although the sample size is not large in this study, a model experiment by the switching housing environment will provide us a unique evidence (this cannot be obtained in the industrial setting).

Overall, the manuscript provides valuable insights into the effects of different housing systems on egg quality traits. The study's findings have implications for the egg industry and provide a foundation for future research in this area.

Response 3:

Thank you very much for giving us warm messages.

Submission Date

09 April 2023

Date of this review

26 Apr 2023 02:07:07

Reviewer 3 Report

The main objection to this article concerns:

1.       No repetitions in the experiment scheme

2.       Too small a group of hens

Drawing conclusions from such small research groups should not take place!

Numerous factors that affect the body can change the tested parameters.

The lack of repetitions in the experiment does not allow to properly evaluate the results obtained!

For this reason, I do not recommend the article for publication.

Other comments

1.       Data on ethics committee approval for the experiment are missing

2.       There is no clear research hypothesis.

3.       Line 88-89 - the sentence should be deleted

4.       How BWG was calculated, was mortality present and included in BWG?

5.       The first sentence in the discussion is redundant

6.       Research conclusions written too briefly - this section should be rewritten

7.       English needs correction

minor editing required

Author Response

We appreciate it for your valuable comments to improve our manuscript. 

animals

Housing systems affect eggshell lightness and free amino acid contents of egg albumen in Tosa-jidori chickens

Nonoka Kawamura 1, Masahiro Takaya 1,2, Hideaki Hayashi 3 and Tatsuhiko Goto 1,4,*

<Reviewer 3>

Comments and Suggestions for Authors

The main objection to this article concerns:

  1. No repetitions in the experiment scheme
  2. Too small a group of hens

Drawing conclusions from such small research groups should not take place!

Numerous factors that affect the body can change the tested parameters.

The lack of repetitions in the experiment does not allow to properly evaluate the results obtained!

For this reason, I do not recommend the article for publication.

Response 1:

In this study, a model study using the Tosa-jidori was conducted to test the effects of deep litter on egg traits. It was difficult to increase the sample size because egg samples were not collected in the industrial layer farm. Although the sample size is not large in this study, a model experiment by the switching housing environment will provide us a unique evidence (this cannot be obtained in the industrial setting).

The aim of this study is to collect robust evidence how the deep litter affects egg traits under the experimental setting. Because of this purpose, the switching the housing system will provide us important knowledge in the future poultry science using the Tosa-jidori model. The evidence using the switching housing system will be more likely to be the result of repetitions.

Other comments

  1. Data on ethics committee approval for the experiment are missing

Response 2:

We have already mentioned in “Institutional Review Board Statement”.

This study was approved by the Experimental Animal Committee of the Obihiro University of Agriculture and Veterinary Medicine (authorization number 18–15).

  1. There is no clear research hypothesis.

Response 3:

Actually, relocation (switching the housing system) is not usual in the layer industry. We would like to collect experimental evidence using the Tosa-jidori model. We added the description as below (L92-98).

To investigate the effect of the housing system, we adopted switching the housing systems (cage and litter) using the Tosa-jidori model. The main reason for the switching experiment is to determine the effect of the housing systems experimentally. If the housing system has some strong effects on egg quality traits, the significant effects on egg traits can be seen in both the first and second stages (2 months each) after switching the housing systems.

  1. Line 88-89 - the sentence should be deleted

Response 4:

We would like to place this sentence.

  1. How BWG was calculated, was mortality present and included in BWG?

Response 5:

BWG was calculated from BW at 7 wk minus BW at 0 wk for the first stage and 14 wk minus BW at 7 wk for the second stage. This method has already mentioned (L150-154).

Mortality was not included in BWG.

  1. The first sentence in the discussion is redundant

Response 6:

The sentence was shorter than the previous one (L341-348).

  1. Research conclusions written too briefly - this section should be rewritten

Response 7:

We rewrote the section (L446-454).

  1. English needs correction

Response 8:

Our manuscript has checked by grammar editing.

Comments on the Quality of English Language

minor editing required

Response 9:

Our manuscript re-checked by grammar editing.

Submission Date

09 April 2023

Date of this review

09 May 2023 11:45:29

Round 2

Reviewer 1 Report

Dear Authors,

It may be more appropriate to make a different article with other data without egg quality characteristics of this study.

Changes in egg quality characteristics are not given in Table 1.

Your explanations about Reference 11 are not sufficient.

 Best regards,

Author Response

Thank you very much. Since we revised our manuscript, would you re-evaluate it? 

animals

Housing systems affect eggshell lightness and free amino acid contents of egg albumen in Tosa-jidori chickens

Nonoka Kawamura 1, Masahiro Takaya 1,2, Hideaki Hayashi 3 and Tatsuhiko Goto 1,4,*

<Reviewer 1>  2nd round

Dear Authors,

It may be more appropriate to make a different article with other data without egg quality characteristics of this study.

Response 1:

We would like to make one research paper including all the data as shown by this version of the manuscript.

Changes in egg quality characteristics are not given in Table 1.

Response 2:

We added the results of the index traits in Table S1 and in L271.

Your explanations about Reference 11 are not sufficient.

Response 3:

In the FAOSTAT, we can move to (1) Rankings; (2) Production, Countries by commodity. By setting “Hen eggs in shell, fresh”, we can see figure of “Top 10 country production of hen eggs in shell, fresh in 2021”. In addition, we can download the csv file from FAOSTAT. The data is shown below.

Area

Element

Item

Year

Unit

Value

Ranking

China, mainland

Production

Hen eggs in shell, fresh

2021

tonnes

2.93E+07

1

India

Production

Hen eggs in shell, fresh

2021

tonnes

6710000

2

United States of America

Production

Hen eggs in shell, fresh

2021

tonnes

6643722

3

Indonesia

Production

Hen eggs in shell, fresh

2021

tonnes

5155998

4

Brazil

Production

Hen eggs in shell, fresh

2021

tonnes

3317193

5

Mexico

Production

Hen eggs in shell, fresh

2021

tonnes

3046910

6

Japan

Production

Hen eggs in shell, fresh

2021

tonnes

2574255

7

Russian Federation

Production

Hen eggs in shell, fresh

2021

tonnes

2496384

8

Türkiye

Production

Hen eggs in shell, fresh

2021

tonnes

1206099

9

Colombia

Production

Hen eggs in shell, fresh

2021

tonnes

1021735

10

We added more information (L61) as below: Japanese layer industry produces a large number of eggs, and the production quantity (approximately 2.57 million tonnes based on “hen eggs in shell, fresh” in 2021) is the seventh largest in the world after China, India, the United States, Indonesia, Brazil, and Mexico [11].

 Best regards,

Submission Date

09 April 2023

Date of this review

24 May 2023 21:13:13

Reviewer 3 Report

The article can be accepted, however, since so few birds were used in the research, and taking into account the authors' responses, it should be added to the title of the work - preliminary research

Author Response

Thank you very much. Since we revised our manuscript, would you re-evaluate it? 

animals

Housing systems affect eggshell lightness and free amino acid contents of egg albumen in Tosa-jidori chickens

Nonoka Kawamura 1, Masahiro Takaya 1,2, Hideaki Hayashi 3 and Tatsuhiko Goto 1,4,*

<Reviewer 3>  2nd round

Comments and Suggestions for Authors

The article can be accepted, however, since so few birds were used in the research, and taking into account the authors' responses, it should be added to the title of the work - preliminary research

Response 1:

Thank you very much for your evaluation. According to your comment, we added “preliminary research” in the title.

Submission Date

09 April 2023

Date of this review

26 May 2023 10:17:15

Round 3

Reviewer 1 Report

Dear Author,

Thank you for the corrections.

Best tegards,